# Surveillance Quality Indicators Highlight the Need for Improving Tuberculosis Diagnostics and Monitoring in a Hyperendemic Area of the Brazilian Amazon Region

**DOI:** 10.3390/tropicalmed7080165

**Published:** 2022-08-03

**Authors:** Juliana Conceição Dias Garcez, Daniele Melo Sardinha, Emilyn Costa Conceição, Gabriel Fazzi Costa, Ianny Ferreira Raiol Sousa, Cristal Ribeiro Mesquita, Wellington Caldas do Carmo, Yan Corrêa Rodrigues, Luana Nepomuceno Gondim Costa Lima, Karla Valéria Batista Lima

**Affiliations:** 1Program in Parasitic Biology in the Amazon Region (PPGBPA), State University of Pará (UEPA), Tv. Perebebuí, 2623-Marco, Belém 66087-662, PA, Brazil; danielle-vianna20@hotmail.com (D.M.S.); gabrielfazzi@gmail.com (G.F.C.); raiolianny@hotmail.com (I.F.R.S.); cristalmesquita@yahoo.com.br (C.R.M.); luanalima@iec.gov.br (L.N.G.C.L.); 2Bacteriology and Mycology Section, Evandro Chagas Institute (SABMI/IEC), Health Surveillance Secretariat, Ministry of Health, Ananindeua 67030-000, PA, Brazil; yan.13@hotmail.com; 3Department of Science and Innovation, National Research Foundation Centre of Excellence for Biomedical Tuberculosis Research, South African Medical Research Council Centre for Tuberculosis Research, Division of Molecular Biology and Human Genetics, Faculty of Medicine and Health Sciences, Stellenbosch University, Cape Town 8000, South Africa; emilyncosta@gmail.com; 4Secretaria Municipal de Saúde de Ananindeua, Ananindeua 67000-000, PA, Brazil; enfwellingtoncaldas@gmail.com

**Keywords:** tuberculosis, epidemiology, surveillance, quality indicators, brazilian Amazon

## Abstract

The city of Ananindeua, State of Pará, Brazil, is a hyperendemic area for tuberculosis (TB). The present study describes the population characteristics and epidemiological indicators of TB cases from Ananindeua, from 2018 to 2020. The TB cases were screened from the Municipal Health Department of Ananindeua database, and the secondary data were obtained from the Brazilian Notifiable Diseases Information System (SINAN). A high percentage of cases did not undergo a rapid molecular test (74.9%) or culture (84.8%) for diagnosis of TB; a chest X-ray examination for diagnosis of TB was performed in 74.47% of new cases. The SINAN form data was incomplete on susceptibility test results (<0.01–92.7). Sputum smear microscopy for monitoring treatment was recorded in the follow-up form in 34.3% and after the 6th month in 61.1% of cases. The cure rate (60.31%) was below the recommendation by the Brazilian Ministry of Health. The quality indicators showed many weaknesses: (I) lack of availability of smear microscopy as a diagnostic test in a hyper-endemic area; (II) low availability of specific exams such as culture and rapid molecular test (RMT); (III) low adherence to smear microscopy to monitor the evolution of cases during treatment; (IV) absence of drug susceptibility test data; (V) failure to fill in essential variables for TB surveillance.

## 1. Introduction

Fighting and controlling tuberculosis (TB), a serious public health problem, is still a challenge in developing countries, as the cases are directly associated with health determinants and conditions, affecting the population with low income and schooling, poor housing conditions, as well as difficulties in accessing health care facilities [1,2].

The Brazilian TB surveillance program relies on the completion of the notification form of the Notifiable Diseases Information System (SINAN) by health professionals in the primary healthcare facility, who offer immediate treatment, evaluation of contacts, and follow-up of the patient until the outcome [3,4].

Epidemiological indicators facilitate the determination of how the disease behaves in a region based on the evaluation of socio-demographic as well as clinical data of the affected individuals, incidence, and lethality and mortality, monitoring the occurrences as a possibility of changing the epidemiological pattern. Surveillance quality indicators, on the other hand, seek to ensure greater efficiency in laboratory diagnosis, treatment, and follow-up until cure and control of new cases [3,5,6,7].

According to the World Health Organization (WHO) report, the TB incidence in the Region of the Americas is slowly increasing, owing to an upward trend in Brazil since 2016. In this context, out of the three global lists of high-burden countries for TB, Brazil belongs to the TB and TB/HIV lists [8], with a national TB average incidence in 2020 of 32.4/100,000 inhabitants [9]. In 2020, there was a 14.3% drop in TB case reporting compared to 2019, which might be influenced by the COVID-19 pandemic [10].

Among the five main Brazilian regions, the North region harbored the highest TB incidence rate in 2020 (46.1/100,000 inhabitants), and, among the 27 states of Brazil, the state of Pará (PA) was ranked at the 5th position (47.1/100,000 inhabitants) on the TB incidence index. The municipality of Ananindeua (PA), with a TB incidence of 89.3/100,000 inhabitants in 2020, characterizes TB hyper-endemicity [11].

In this alarming context, we aimed to describe the population characteristics and epidemiological indicators of TB cases from the city of Ananindeua (PA), Brazil, from 2018 to 2020, by comparing the new cases versus relapse cases.

## 2. Materials and Methods

### 2.1. Study Design and Location

This is a cross-sectional epidemiological research based on secondary data from SINAN (http://portalsinan.saude.gov.br/tuberculose, accessed on 21 March 2022) notifications [12] from 1 January 2018 to 31 December 2020 that was analyzed for TB cases in the municipality of Ananindeua, state of Pará, Brazil. This study followed the recommendations of The Strengthening the Reporting of Observational Studies in Epidemiology (STROBE) Statement: guidelines for reporting observational studies [13].

The municipality of Ananindeua represents the second most populous municipality in the state of Pará, with an estimate of 540,410 inhabitants for 2021 in a territorial extension of 190,451 km, as shown in Figure 1. It is located in the 6th micro-region of the state of Pará, bordering the municipalities of Belém and Marituba, with its coverage of basic sanitation in 2010 determined to be 55.1% [14].

### 2.2. Tuberculosis Case Selection

The cases were selected from the analysis of the database provided by the Municipal Health Department of Ananindeua. We included all residents of the Ananindeua who were not transferred to other municipalities. Cases that did not meet the inclusion criteria, duplicate cases, and those with dubious identification were excluded.

The criteria for defining cases were in accordance with the Brazilian Ministry of Health: (i) laboratory criteria—every case that, regardless of the clinical form, presents at least one positive sample from smear microscopy, the culture exam, or the positive rapid molecular test; (ii) clinical-epidemiological criterion—every case that did not meet the laboratory confirmation criteria described above but received a diagnosis of active TB. This definition takes into account clinical-epidemiological data associated with the evaluation of other complementary exams such as imaging and histological, among others [15]. Population screening for the study was performed according to Figure 2.

### 2.3. Data Collection and Analysis

The variables selected for the study were extracted from SINAN, specifically from the TB case notification form and follow-up that comprises a total number of 24 indicators, but in order to meet the objectives of the study, 17 indicators were selected related to epidemiological issues, clinical forms, types of diseases and associated diseases, smear microscopy, X-ray, closure situation, treatment directly observed (DOT), molecular test, histopathological, and culture performance.

The calculation of incidence (number of cases/by estimated population of the year x 100,000) and mortality (number of deaths/by number of cases ×100) were performed to describe the epidemiological indicators.

We performed statistical analysis using the statistical program Statistical Package for the Social Sciences 20 (IBM® SPSS®, Armonk, New York, U.S). The chi-square statistical tests of adherence (independence) were used for the profile of cases and surveillance quality indicators, and the chi-square test (2 × 2 table) and G test (Contingency Table L × C) were used for the values less than five (*p* < 5). The odds ratio for significant variables (<0.05) was also performed to associate significant variables between new and relapsed cases.

### 2.4. Ethical Aspects

The database was made available by the Municipal Health Department of Ananindeua (SESAU) through the authorization letter for the use of data issued by SESAU, and it appears ethically approved under No. 4.172.679. Respecting resolution 466/12, which establishes criteria for research with human beings, we preserved the confidentiality and security of the participants in accordance with the Declaration of Helsinki. The database was only manipulated by the researchers, thereby minimizing the risk of data leakage as well as enabling the exclusion of the patients’ names. The Free and Informed Consent Term (ICF) was waived by the Research Ethics Committee, as it is secondary data referring to a 3-year retrospective cohort.

## 3. Results

### 3.1. Epidemiological Indicators

The distribution of 1434 TB cases per year was: 428 in 2018 (no death by TB), 528 in 2019 (three deaths by TB), and 478 in 2020 (six deaths by TB). From 2018 to 2020, we observed a linear trend line that indicates the TB incidence increase; regarding the mortality, it ranged between 0 to 1.26% (Figure 3).

The profile characteristics demonstrated that the 1434 participants are majority male (60.60%; 869), parda (76.01%; 1090), and aged between 20 and 59 years (74.33%; 1066); almost 40% (550) of the cases presented from incomplete high school to complete higher education. There was a negligence in this variable regarding filling out the form, since the “ignored” represented <0.001–18.76% of the participants. The type of entry “new cases” and the “pulmonary” form were the most reported. Regarding the extrapulmonary forms, they were pleural (5.3%) and peripheral ganglionic (3.0%). On the chest X-ray, the “suspect” was more predominant (<0.001–69.04%) (Table 1).

### 3.2. Surveillance Quality Indicators

The sputum smear test performed in the initial consultation allowed the identification of 56.6% of the bacilliferous cases. A high percentage of cases (29.4%; 421) did not undergo the examination. Regarding the rapid molecular test indicator, most did not perform it (74.9%; 1074), which reflects the low quality of surveillance; of those who performed it, they were detected to be susceptible to rifampicin (<0.001–19.9%; 285). Regarding the performance of the susceptibility test, most are not completed (<0.001–92.7%; 1.329).

Regarding the culture exam, the reference standard diagnosis, it was not performed in almost all cases (84.8%; 1216). Regarding sputum smear microscopy for monitoring treatment, it was observed that the smear microscopy was not recorded in the follow-up form in 34.3% (492) of cases and after the 6th month in 61.1% (876) of cases (Table 2).

### 3.3. New Cases Versus Relapses

Out of 1432 cases, 72 (5.02%) were relapse TB cases, among which two variables were associated to relapse: education (“Incomplete Elementary School: 5th to 8th series” (0.009–26.76%- OR 2.1558- CI 1.2454–3.7315)) and failure to perform the X-ray (0.0028–38.03%- OR 1.7906-CI 1.0906–2.9399). On the other hand, the variable chest X-ray “Suspect” was associated with new cases (0.053–59.15% -OR 0.6021-CI 0.3693–0.9815) (Table 3).

The group comparison amongst new cases and relapses regarding the conditions of comorbidities, behaviours, risk groups, and disease outcomes demonstrated that two groups were associated with relapse cases: people deprived of liberty (0.002–12.68%-OR 3.3220-CI 1.5656–7.0486) and people for whom DOT was not performed (0.001–88.73%-OR 3.5913 -CI 1.7043–7.5676) (Table 4).

## 4. Discussion

Our study demonstrated that the socio-demographic characteristics of 1432 TB cases diagnosed in Ananindeua (PA) (men, adults (20–39 years old), economically active age group, subsequent to patients belonging to the age group of 40 to 59 years, especially males with incomplete elementary school education) are in agreement with a previous study in the same municipality from 2010 to 2014, which presented similar profiles: male (59.54%), parda race (74.69%; 944), incomplete elementary level education (39.44%; 304), of the adult age group (19 to 59 years) (87.32%; 1048) [16].

Regarding the laboratorial tests, the performance of microscopy in only 79.1% of suspected cases emphasizes the need for greater attention and supervision of the service offered. All primary care units should offer smear microscopy for TB diagnosis at the time the patient seeks the health unit and after instructions for sample collection at dawn. The Brazilian Ministry of Health recommends laboratory diagnosis in 100% of investigated cases, with the culture exam recommended for at least 80% of reported cases [17,18].

A quality study on TB surveillance in Brazil in 2013 showed that primary health care was, at the time, successfully incorporating the management of new smear-positive TB cases. Primary health care achieved better operational indicators than secondary or tertiary levels [19].

As Ananindeua is a hyper-endemic area, the Ministry of Health recommends that all cases undergo a rapid molecular test (RMT) for rapid diagnosis and investigation of rifampicin resistance [3]. In this study, RMT was performed in less than 25% of the population, and the culture exam was performed in approximately 15% of the population. Regarding the drug susceptibility test (DST) data, there was a lack of notification for more than 90% of the cases.

The follow-up stage is essential for the patient’s cure as well as to obtain the variables related to each stage, such as collection of culture exams in the treatment process, RMT, DST, and the outcome (closing situation). For data qualification, there are criteria considering three different situations for treatment time: <150 days (cure, multidrug-resistant (MDR)-TB, abandonment, or transfer), between 150 and 270 days, and greater than 270 days (relapse or re-entry after abandonment). Cure is defined when the patient completes the period of 180 days of treatment, regardless of bacteriological confirmation [20].

We observed many cases (256) with incomplete information in the closing situation stage, which reflects the low quality of TB surveillance and control in the municipality. Other studies also highlighted this problem, demonstrating that the completeness of the initial variables investigation’s introduction reached the recommendations of the Brazilian Ministry of Health; however, there is a lack in completeness in the follow-up stages. This scenario suggests a deficiency of a global understanding by the health professional who attends the patients and reports their information. In this case, the notification system is observed as a laborious bureaucracy process, and the responsibility is transferred to the municipal surveillance to investigate and complete SINAN database; however, due to the high demand for numbers of TB cases throughout the municipality, the analysis of the database is neglected by municipal and state surveillance, with most cases remaining unsolved and without disease control [7,21].

Upon the analysis of case closure, the cure rate was much lower than expected by the Brazilian Ministry of Health, with many cases not filling out the form. The number of cases with no outcome is also worrying. Farias et al. [22]. highlight that the fragility in these indicators reflects the lack of control of the disease, thus increasing new cases and mortality, which still implies difficulties in the maintenance of the disease, a greater possibility of drug-resistant TB, and higher costs with the disease and human suffering, and shows that surveillance also neglects the disease, which is the main factor in the lack of control of the endemic.

A survey in Brazil associated the highest incidences of TB and clusters of cases with vulnerabilities and failures in the development of epidemiological and operational indicators of surveillance in relation to TB in the regions [23].

The continuous evaluation of surveillance indicators, based on specific methods for each disease, such as TB indicators, that highlight epidemiological and operational factors, is especially essential in middle- and low-income countries as a strategy for combating the TB epidemic; the evaluation of operational indicators that include more attributes than the surveillance quality indicators shows the capacity of local surveillance to notify, investigate, and guarantee the breakage of the chain of transmission and cure of the patient. It also concludes that in order to combat the disease, in addition to operational research, it is also necessary to invest in vaccine research, laboratory diagnosis, and treatment [24].

In an integrative review of the evaluation of the performance of the TB control program from Brazilian and Spanish studies, the authors concluded that the unpleasant results reflect the complexity of the evaluation of programs that involve multiple activities and diverse actors, pointing to the need for the integration of epidemiological and operational indicators in TB [25].

This study also showed several factors associated with cases of recurrence, such as: (i) low education, (ii) deprived of liberty, and (iii) non-performance of the directly observed treatment. A population-based, longitudinal retrospective study carried out in Barcelona, Spain identified as factors associated with TB recurrence: male gender, being an immigrant, being an injecting drug user, being HIV positive, smoking, drinking, being in prison, and having pulmonary and extrapulmonary TB [26]. This study corroborates only those deprived of liberty, who were associated with relapse in this study.

Another study of factors associated with relapse in England and Wales identified only HIV co-infection and belonging to a South Asian ethnic group [27]. A meta-analysis linked alcohol consumption to the risk of TB recurrence, with it being the main contributor [28]. A large study in London from 2002 to 2015 linked TB relapses as well as disease reinfections to social factors and surveillance weaknesses such as low sensitivity of respiratory symptomatic case reports and inadequate investigation of new cases, and it found that in these London locations there is no control of the disease and, thus, transmission is taking place [29].

## 5. Conclusions

This study analyzed the epidemiological indicators and showed that the incidence of TB in the Ananindeua-Pará municipality is much higher than the state and national average, characterizing a hyperendemic region. The cases are concentrated in male individuals, aged between 20 and 59 years. Almost 40% of the cases (550) had an education equal to or higher than incomplete high school, evidencing the occurrence of the disease in a more educated population.

The quality indicators showed many weaknesses, ranging from diagnosis to case monitoring and outcome recording: (i) lack of availability of smear microscopy as a diagnostic test in a hyper-endemic area; (ii) low availability of more specific exams such as culture and RMT; (iii) low adherence to smear microscopy to monitor the evolution of cases during treatment; (iv) absence of DST data; (v) failure to fill in essential variables in forms for TB surveillance. We also identified the factors associated with relapse: incomplete 5th to 8th grade schooling, non-performance of chest X-rays, deprivation of liberty, and directly observed treatment not performed. Further studies aiming to evaluate the progression on the deficient indicators highlighted in this investigation, including interventions to prevention and control as well as investigation of transmissions chains by applying a molecular epidemiology-based approach, are necessary to assist and strengthen TB vigilance in the city of Ananindeua.

## Figures and Tables

**Figure 1 tropicalmed-07-00165-f001:**
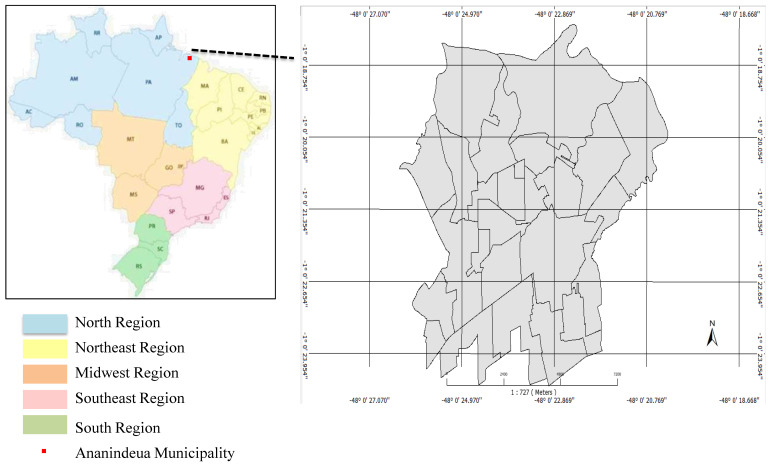
Municipality of Anandindeua/Pará-2022. Source: TerraView/IEC/UEPA, 2022.

**Figure 2 tropicalmed-07-00165-f002:**
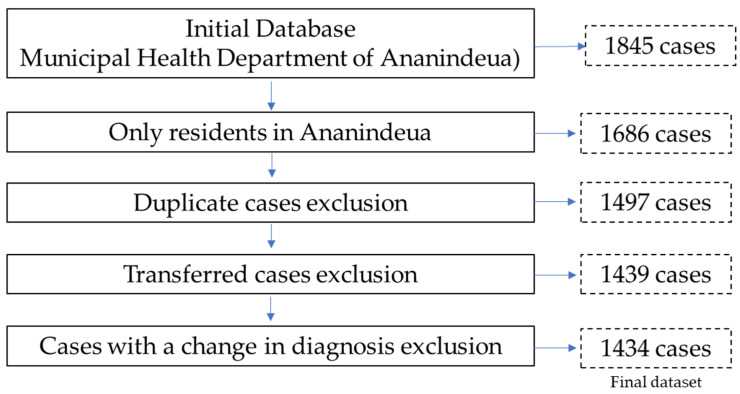
Population screening for the study. Tuberculosis, Ananindeua/Pará, 2018 to 2022. Data source: SINAN.

**Figure 3 tropicalmed-07-00165-f003:**
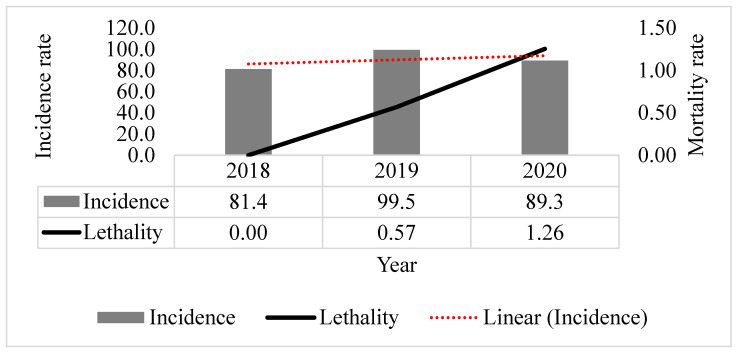
Tuberculosis incidence (per 100,000 inhabitants) and mortality rates in Ananindeua, Pará from 2018 to 2020. Data source: SINAN.

**Table 1 tropicalmed-07-00165-t001:** Profile of tuberculosis cases in Ananindeua, Pará from 2018 to 2020.

Variable	Total 1434	%	*p*-Value
Sex			
Female	565	39.40	<0.001
Male	869	60.60	
Age Group			
<1	21	1.46	<0.001
1 to 19	155	10.81	
20 to 39	628	43.79	
40 to 59	438	30.54	
>60	192	13.39	
Race			
White	163	11.37	<0.001
Black	139	9.69	
Yellow	4	0.28	
Parda	1090	76.01	
Indigenous	1	0.07	
Ignored	37	2.58	
Education			
Illiterate	17	1.19	<0.001
1st to 4th incomplete elementary school	143	9.97	
2nd to 4th complete elementary school	66	4.60	
5th to 8th incomplete elementary school	222	15.48	
Complete elementary school	100	6.97	
Incomplete high school	165	11.51	
Complete high school	286	19.94	
Incomplete undergraduate education	46	3.21	
Complete undergraduate education	53	3.70	
Ignored	269	18.76	
Data does not apply	67	4.67	
Entry Type			
New case	1242	86.61	<0.001
Relapse	71	4.95	
Re-entry after abandonment	35	2.44	
Uncrown	5	0.35	
Transference	81	5.65	
Tuberculosis Type			
Pulmonary	1275	88.91	<0.001
Extrapulmonary	141	9.83	
Pulmonary + Extrapulmonary	18	1.26	
Chest X-ray			
Suspect	990	69.04	<0.001
Normal	42	2.93	
Other pathology	11	0.77	
Not performed	391	27.26	
Extrapulmonary Type			
Pleural	76	5.30	
Others	12	0.84	
Peripheral ganglion	43	3.00	
Bone	10	0.70	<0.001
Ocular	1	0.07	
Miliary	9	0.63	
Meningoencephalocele	4	0.28	
Cutaneous	2	0.14	
Laryngeal	2	0.14	

Data source: SINAN.

**Table 2 tropicalmed-07-00165-t002:** Surveillance quality indicators. Tuberculosis cases in Ananindeua, Pará from 2018 to 2020.

Surveillance Quality Indicators	N	%	*p*-Value
Sputum Smear			
Positive	812	56.6	<0.001
Negative	274	19.1	
Not performed	299	20.9	
Does not apply	49	3.4	
Rapid Molecular Test			
Detectable-Rifampicin susceptible	285	19.9	<0.001
Detectable-Rifampicin resistant	13	0.9	
Undatable	24	1.7	
Inconclusive	34	2.4	
Not performed	1074	74.9	
Not filled	4	0.3	
Drug Susceptible Test			
Not filled	1329	92.7	<0.001
Resistant to other first-line drugs	1	0.1	
Susceptible	6	0.4	
Ongoing	7	0.5	
Not performed	91	6.3	
Culture			
Positive	103	7.2	<0.001
Negative	58	4.0	
Ongoing	57	4.0	
Not performed	1216	84.8	
Sputum Baciloscopy (6 Months)			
Record not completed	492	34.3	<0.001
Negative	378	26.4	
Not performed	421	29.4	
Not applied	143	10.0	
Sputum Baciloscopy (After 6 Months)			
Record not completed	876	61.1	<0.001
Negative	145	10.1	
Not performed	270	18.8	
Not applied	143	10.0	
Outcome Situation			
Record not completed	289	20.2	<0.001
Cure	849	59.2	
Primary Abandonment	1	0.1	
Abandonment	132	9.2	
Death by TB	9	0.6	
Death by other causes	30	2.1	
Transference	101	7.0	
Diagnostic change	13	0.9	
TB-DR	8	0.6	
Mudança de Esquema	2	0.1	

Data source: SINAN.

**Table 3 tropicalmed-07-00165-t003:** New cases versus relapses profile in Ananindeua, Pará from 2018 to 2020.

Variables	New Cases (1242)	%	Relapses (71)	%	* *p*-Value	^#^ OR	^&^ CI
Sex					0.648		
Female	498	40.10	26	36.62			
Male	744	59.90	45	63.38			
Age Group							
<1	20	1.61	0	0.00	0.520		
1 to 19	138	11.11	4	5.63	0.123		
20 to 39	530	42.67	35	49.30	0.331		
40 to 59	380	30.60	23	32.39	0.851		
>60	174	14.01	9	12.68	0.889		
Race							
White	147	11.84	7	9.86	0.754		
Black	117	9.42	8	11.27	0.758		
Yellow	3	0.24	0	0.00	0.482		
Parda	944	76.01	53	74.65	0.906		
Indigenous	1	0.08	0	0.00	0.207		
Ignored	30	2.42	3	4.23	0.597		
Education							
Illiterate	15	1.21	0	0.00	0.702		
Incomplete Elementary School: 1st to 4th series	124	9.98	9	12.68	0.597		
Complete Elementary School: 2nd to 4th series	49	3.95	6	8.45	0.164		
Incomplete Elementary School: 5th to 8th series	180	14.49	19	26.76	0.009	2.1558	1.2454–3.7315
Complete Elementary School	92	7.41	5	7.04	0.905		
Incomplete High School	150	12.08	6	8.45	0.465		
Complete High School	252	20.29	12	16.90	0.589		
Incomplete Undergrad	40	3.22	2	2.82	0.876		
Complete Undergrad	51	4.11	2	2.82	0.818		
Ignored	266	21.42	10	14.08	0.185		
Not applied	23	1.85	0	0.00	0.435		
Form							
Pulmonary	1116	89.86	66	92.96	0.519		
Extrapulmonary	112	9.02	5	7.04	0.723		
Pulmonary + Extrapulmonary	14	1.13	0	0.00	0.746		
Chest X-ray							
Suspect	878	70.69	42	59.15	0.053	0.6021	0.3693–0.9815
Normal	38	3.06	1	1.41	0.646		
Other pathology	9	0.72	1	1.41	0.954		
Not performed	317	25.52	27	38.03	0.028	1.7906	1.0906–2.9399
Extrapulmonary Types							
Pleural	63	5.07	1	1.41	0.215		
Other	9	0.72	0	0.00	0.984		
Peripheral ganglion	34	2.74	3	4.23	0.722		
Bone	7	0.56	0	0.00	0.846		
Ocular	0	0.00	1	1.41	0.207		
Miliary	8	0.64	0	0.00	0.918		
Meningoencephalic	1	0.08	0	0.00	0.207		
Cutaneous	2	0.16	0	0.00	0.360		
Laryngeal	2	0.16	0	0.00	0.360		

Source: SINAN. * G test, # Chi-square Odds ratio with ^&^ confidence interval.

**Table 4 tropicalmed-07-00165-t004:** New cases versus relapses (conditions and outcome) in Ananindeua, Pará from 2018 to 2020.

Variables	New Cases (1242)	%	Relapses (71)	%	* *p*-Value	^#^ OR	^&^ CI
HIV	106	8.53	8	11.27	0.563		
Diabetes	160	12.88	6	8.45	0.363		
Mental disease	21	1.69	2	2.82	0.817		
Alcoholism	126	10.14	11	15.49	0.217		
Others	104	8.37	5	7.04	0.860		
Drugs	78	6.28	8	11.27	0.192		
Smoking	137	11.03	9	12.68	0.818		
Deprived of liberty	52	4.19	9	12.68	0.002	3.3220	1.5656–7.0486
Street situation	7	0.56	0	0.00	0.846		
Immigrants	4	0.32	0	0.00	0.588		
Health professionals	23	1.85	0	0.00	0.435		
^+^ DOT not performed	853	68.68	63	88.73	0.001	3.5913	1.7043–7.5676
Cure	749	60.31	39	54.93	0.438		
Abandonment	110	8.86	9	12.68	0.380		
Death by TB	9	0.72	0	0.00	0.984		
TB-DR	5	0.40	1	1.41	0.768		

Data source: SINAN. + Directly Observed Treatment, * G-Test or Chi-square, ^#^Odds ratio, ^&^ confidence interval.

## Data Availability

All relevant data is presented within the manuscript.

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
