# Peer review of "Surveillance Quality Indicators Highlight the Need for Improving Tuberculosis Diagnostics and Monitoring in a Hyperendemic Area of the Brazilian Amazon Region"

_tropicalmed, 2022, doi:10.3390/tropicalmed7080165_

Round 1

Reviewer 1 Report

Thanks for recommending me as a reviewer. In this paper, the authors described the demographic characteristics and epidemiologic indicators of tuberculosis cases in Ananindeua from 2018 to 2020. Tuberculosis cases were screened in the database of the Municipal Health Department in Ananindeua, with secondary data. If the authors complete the revision, the quality of the study will be further improved.

1. The introduction section is well written. If the authors describe the trends of prior research on Improving Tuberculosis Diagnostics and Monitoring in a Hyperendemic Area in more detail in the introduction section, it can help readers understand.

2. Table 1 and 2: The authors set the significance level as <0.01. However, in general, it is appropriate to express the significance level as <0.05. In my opinion, it is better to indicate the significance level (ex. 0.031) in the tables.

3. It is recommended that the last paragraph of the Conclusion section be made as an implication for future research.

Reviewer 2 Report

The authors submitted an extensive report about the deficiency in diagnosis, testing resources, and follow-up records. It is unfortunate that the city of Ananindieua is struggling to upgrade its TB surveillance and treatment strategy. 

Major comments:

Table 2 reflects the negligence in quality indicators for diagnosis. That is a huge number of cases that missed sputum smear, molecular, and drug tests. It is a poor surveillance system. Is it not alarming to the authorities due to the low mortality rate?

Table 4 showed the high % of non-DOT treated cases. What could be the reason? Lack of resources from the WHO or governmental organization? It will keep close contact and a higher risk of contracting the pathogen.

Minor comments:

Line 54: typo/space ..2021 World Health Org...

Line 96 : typo...total number of participants....

Line 167: Keep uniformity in the table nomenclature. The authors used male/female in Table 2 but it was mentioned as feminine/masculine in table 3.

Line 222: Typo ...cure rate was..

Reviewer 3 Report

Article well written.

I have not any suggestions.

Author Response

We thank the reviewer comments.